# FANTASTIC DNN-CLASSIFIER IDENTIFICATION WITHOUT TESTING DATASET

## ABSTRACT

Deep Neural Networks (DNNs) are trained, validated, and tested with an example datasets. Finally, performance of the selected DNN is tested using a test dataset. This testing method treats the DNN as a black-box and doesn't attempt to understand its characteristics. On the other hand, many theoretical and empirical studies have used complexity measures for estimating generalization phenomenon using training dataset, with rare exceptions. To the best of our knowledge, no method exists for estimating test-accuracy (not generalization) without any testing dataset. We propose a method for estimating *test accuracy bounds* of a given DNN *without any test dataset*. Assuming that a DNN is a composition of a feature extractor and a classifier, we propose and evaluate a method for estimating their qualities. The first step of the proposed method is generation of one (input) prototype vector for each class. Then using these seed prototypes, $(k-1)$ *core* prototypes are generated for each class. These prototypes are our data for evaluating qualities of the feature extractor and classifier as well as estimating test accuracy of the given DNN. We have empirically evaluated the proposed method for ResNet18 trained with CIFAR10 and CIFAR100 datasets.

## 1 INTRODUCTION

Often Machine Learning (ML) models are trained, validated, and tested with example data. A significant part of the data is used to estimate the parameters of a model, next validation data is used for hyperparameter and model selection, and finally the test data is used to evaluate performance of the selected model (Raschka, 2020; Zheng et al., 2019). *Deep Neural Networks* (DNNs) follow a similar method, where training examples are used to estimate parameters of a DNN. For a given example dataset, different model-architectures are trained and using a validation dataset a model is selected. Modern DNN models have hyper-parameters and their good values are selected using validation datasets as well. Finally, performance of the selected DNN (network parameters and hyper-parameters) is tested using a test dataset.

Even when training, validation, and test datasets are coming from the same distribution or created by partitioning a larger dataset, training accuracy is often much higher than test accuracy. For example, ResNet18 (He et al., 2015) trained with CIRAR100 (Krizhevsky et al., b) achieves almost 100% accuracy very quickly, but testing accuracy is between 72 to 75%. This gap between training and test accuracy is viewed as (partial) memorization of (some) examples rather than learning underlying features. A fundamental question is how to find what the network has learned. Can this be estimated without any data?

We propose a method for estimating test accuracy bounds of a given DNN, *without test dataset*. Assuming that a DNN is a composition of a feature extractor and a classifier, we propose and evaluate a method for estimating quality of the feature extractor and the classifier. We propose two metrics for estimating quality of the feature extractor and one metric for the classifiers. These metrics are then used to estimate test accuracy bounds of the given DNN. The first step of the proposed method is generation of one (input) prototype vector for each class (see Algorithm 1). Then using these seed prototypes, $(k-1)$ *core* prototypes are generated for each class (see Algorithm 2). These prototypes are our dataset.

After forward-passing each prototype through the feature extractor, a feature vector is obtained. Repeating this process with all the prototypes, a total of $k^2$ feature vectors, $k$ for each class, are

generated and used to estimate mean values of two proposed metrics and their variances, which are used to produce an estimate of test accuracy of the given DNN. The metric for assessing the classifier, we use its weight vectors. For concise and precise descriptions we introduce the notions that we use in the paper.

## 1.1 NOTATIONS

Let $[n]$ be the set $\{1, 2, \cdots, n\}$. Given, $\mathscr{D} = \{(x_i, y_i) | x_i \in \mathbb{R}^p, y_i \in [k]$ and $i \in [n]\}$, a dataset of $n$ labeled data coming from $k$ classes $C_1$ to $C_k$. Let $\mathscr{D}_l = \{(x_i, l) | (x_i, l) \in \mathscr{D}$ and $l \in [k]\}$ be the set of data in classes $C_l$. Denoting $|\mathscr{D}_l| = n_l$, we have $n = \sum_{i=1}^{k} n_i$.

We consider multiclass classifier functions $f : \mathbb{R}^p \to \mathbb{R}^k$ that maps an input $\mathbf{x} \in \mathbb{R}^p$ to a class $l \in [k]$. For our purposes, we assume $f$ is a composition of a *feature extraction* function $g : \mathbb{R}^p \to \mathbb{R}^q$ and a *feature classification* function $h : \mathbb{R}^q \to \mathbb{R}^k$. Assume that $f$ is differentiable and after training with a dataset $\mathscr{D}$, a parametric function $f(\cdot\,; \theta)$ is obtained. Because $f$ is a composition of $g$ and $h$, training of $f(\cdot\,; \theta)$ produces $g(\cdot; \theta_g)$ and $h(\cdot; \theta_h)$, where $\theta = \theta_g \cup \theta_h$ and $\theta_g \cap \theta_h = \oslash$.

Let $\mathbf{v} = g(\mathbf{x}; \theta_g)$ and $\hat{\mathbf{y}} = h(\mathbf{v}; \theta_h)$, where $\hat{\mathbf{y}}_l = h_l(\mathbf{v}; \theta_h)$ represents the probability $h_l(\mathbf{v}; \theta_h)$ belongs to class $l \in [k]$ such that $\sum_{l=1}^{k} \hat{\mathbf{y}}_l = 1$. Correspondingly, for any given $\mathbf{x}$, which possess a true label $l$, the classifier could generate $k$ possible class assignments and is correct only when $l = \arg\max_k(\hat{\mathbf{y}})$.

*Unless stated otherwise*, we assume that activation functions at the output layer of the classifier $g(\cdot; \theta_g)$ are ReLu, and $h(\cdot; \theta_h)$ is a fully connected one layer neural network, where $\mathbf{W}_{i,:}, i \in [k]$ be the $i$th row of $h(\cdot; \theta_h)$'s weight matrix $\mathbf{W}$. It receives input from a feature extractor $g(\cdot; \theta_g)$, and $h(\cdot; \theta_h)$'s output is using a softmax function (see Eqn.1) to generate one-hot coded outputs.

$$\sigma_l(\mathbf{W}\mathbf{v}) = \frac{e^{(\mathbf{W}_{l,:} \cdot \mathbf{v})}}{\sum_{i=1}^{k} e^{(\mathbf{W}_{i,:} \cdot \mathbf{v})}} \tag{1}$$

**Problem Statement** We are given a trained neural network $f(\cdot; \theta) = h(g(\cdot; \theta_g); \theta_h)$ for the classification of input vectors $\mathbf{x} \in \mathbb{R}^p$ to one of the $k$ classes; that is, we are given the architecture of the network and values of the parameters $\theta$. No training, validation, or testing data is provided. The network uses a one-hot encoding using softmax function. The problem is to estimate test accuracy bounds of the network. *This is a harder problem, because no data is provided to us.*

Our objective is to develop methods for assessing the network's training quality (accuracy). To be more specific, we want to develop metrics that can be used to estimate training quality of $g(\cdot; \theta_g)$ and $h(\cdot; \theta_h)$. We propose and empirically validate two metrics for $g$ and one for $h$. The metric for $h$ is based on the weight vector $\mathbf{W}$ of $h$, and metrics for $g$ are based on the features that $g$ outputs when synthesized vectors are input to $g$. We propose a method for generating *synthesized vectors* and call them *prototype vectors* (see Sec. 3.1) unless stated otherwise.

We propose a method for evaluating a trained network without knowledge of its training data and hyperparameters. Our method involves $(i)$ generation of **prototype data** for each class, $(ii)$ observation of activations of neurons at the output of the feature extraction function $g(\cdot; \theta_g)$ (which is input to the classification function $h(\cdot; \theta_h)$), and $(iii)$ asses quality of the given network using observations from previous step. The last step, assessment of the quality of a trained network without testing data, requires one or more metrics, which are developed here.

**Related Work:** To the best of our knowledge, no work for estimation of testing accuracy of a trained DNN classifier without dataset exists, except in (Dean & Sarkar, 2023) where it was shown that test accuracy is correlated to cosine similarity of between class prototypes, but their prediction was inaccurate. Learning from their limitations, we propose a method for providing lower and upper bounds of test accuracy.

A related and very important problem, generalization, has been studied theoretical and empirical by many. A good summary of these efforts can be found in (Jiang et al., 2019). After extensive evaluation of thousands of trained networks, Jiang et al. (2019) reported that PAC-Bayesian bounds are good predictive measures of generalization and several optimizations are predictive of generalization, but regularization methods are not. Their evaluation method, like ours, doesn't require a dataset, but their objective is *generalization ability* prediction and our objective is *test accuracy* estimation.

**Training Set Reconstruction:** Recent work (Buzaglo et al., 2023; Haim et al., 2022) has successfully reconstructed training examples by optimizing a reconstruction loss function that requires only the parameters of the model and no data. An underlying hypothesis of this work is that neural networks encode specific examples from the training set in their model parameters. We utilize a similar concept to construct '*meta*' training examples by optimizing a cross-entropy loss at the model's output and requiring significantly lower computation time.

**Term Prototype in Other Contexts:** We want the reader to be aware that term prototype we use have no **direct** relation to the training data and **it should not** be confused with or associated with same term has been used in many other contexts (Mensink et al., 2013; Rippel et al., 2015; Snell et al., 2017; Li et al., 2017; Schwartz et al., 2018; Chen et al., 2018; Guerriero et al., 2018; Mustafa et al., 2019).

The main contribution of our work are:

- We propose a method for generating a synthetic dataset from a trained DNN with one-hot coded output. Because each data in the dataset is correctly classified by the DNN with very high probability values, if the DNN is well trained the data represents an input area from where the DNN must have received training examples.

- We have defined one metric for estimating training quality of the classifier and two metrics for estimating training quality of the feature extractor. Recall that we assume that a DNN is a composition of a feature extractor and a classifier. Using the last two metrics we have developed a method for estimating lower and upper bounds for test accuracy of a given DNN and training or testing data.

- We have empirically demonstrated that the proposed method can estimate lower and upper bounds of actual test accuracy of ResNet18 trained with CIFAR-10, and CIFAR-100.

## 2 Characteristics of the feature extracor $h(\cdot; \theta_h)$

An optimal representation of training data minimize variance of intra-class features and maximize inter-class variance (Bengio et al., 2013; Tamaazousti et al., 2020). Inspired by these ideas in these hypotheses, we define training quality metrics using cosine similarity of features of *prototypes generated from trained DNNs* (see Sec 3.1). Let us deal with a much simpler task of defining a metric for a classifier $h(\cdot; \theta_h)$ defined as the last completely-connected layer.

### 2.1 Training Quality of the Classifier

Let $\mathbf{W}_{i,:}$ be the $i$th row of $h(\cdot; \theta_h)$'s weight matrix $\mathbf{W}$. Recall that elements of the input vector $\mathbf{v}$ to $h(\cdot; \theta_h)$ are non-negative, because they are outputs of ReLU functions.

**Definition 2.1** (Well trained Classifier $h(\cdot; \theta_h)$). *A one-hot output classifier $h(\cdot; \theta_h)$ with weight matrix $\mathbf{W}^{k \times q}$ is well trained, if for each class $l$ there exists a feature vector $\mathbf{v}^{(l)}$ such that*

$$e^{(\mathbf{W}_{l,:} \cdot \mathbf{v}^{(l)})} >> e^{(\mathbf{W}_{i,:} \cdot \mathbf{v}^{(l)})} \text{ for all } i \neq l, \text{ and} \tag{2}$$

$$e^{(\mathbf{W}_{i,:} \cdot \mathbf{v}^{(l)})} \leq 1 \text{ for all } i \neq l. \tag{3}$$

For correct classification, the condition 2 is sufficient. The condition 3 is a stronger requirement but enhances the quality of the classifier, and it leads us to the following lemma.

**Lemma 2.1** (Orthogonality of Weight Vectors of $h(\cdot; \theta_h)$). *Weight vectors $\mathbf{W}_{i,:}$ of a well trained $h(\cdot; \theta_h)$ are (almost) orthogonal.*

*Proof.* From the condition 3 of the Def. 2.1 we assume, $e^{(\mathbf{W}_{i,:} \cdot \mathbf{v}^{(l)})} = 1$ for all $i \neq l$. This implies that $(\mathbf{W}_{i,:} \cdot \mathbf{v}^{(l)}) = 0$ for all $i \neq l$, that is, $(k-1)$ weight vectors $\mathbf{W}_{i,:}$ and $i \neq l$ are orthogonal to $\mathbf{v}^{(l)}$.

Now the value of $(\mathbf{W}_{l,:} \cdot \mathbf{v}^{(l)})$ will determine the angle between $\mathbf{W}_{l,:}$ and $\mathbf{v}^{(l)}$. Because of the condition 2 of the Def. 2.1, it is much higher than zero and a maximum of $(\mathbf{W}_{l,:} \cdot \mathbf{v}^{(l)}) = ||(\mathbf{W}_{l,:}|| \, ||\mathbf{v}^{(l)})||$ is obtained, when they are parallel to each other. If maximum is reached then $\mathbf{W}_{l,:}$ is perpendicular to

all other weight vectors. When this maximum is reached for all $l$, the weight vectors $\mathbf{W}$ are mutually orthogonal. Otherwise, they are almost orthogonal. $\qquad\square$

Our empirical studies, reported in Sec. 4, found that weight vectors of ResNet18 trained with CIFAR10 and CIFAR100 have average angles are 89.99 and 89.99 degrees, respectively. An obvious consequence of the Lemma 2.1 is that there exists a set of $k$ feature vectors each of which are (almost) parallel to the corresponding weight vectors, which is stated as a corollary next.

**Corollary 2.1.1** (Parallelity of $\mathbf{W}_{l,:}$ and $\mathbf{v}^{(l)}$)**.** *If a classifier is well trained, for each weight vector* $\mathbf{W}_{l,:}$ *there is one or more feature vectors* $\mathbf{v}^{(l)}$ *that are (almost) parallel to it.*

**Classifier's Quality Metric, $\mathscr{H}_w$,** postulates that weight vectors of a well trained classifier are (almost) orthogonal to each other. A method for its estimation is defined next.

**Definition 2.2** (Estimation of metric $\mathscr{H}_w$)**.** *An estimation of $\mathscr{H}_w$ is calculated by consider a pair-wise combinations of the weight matrix of h, and given by*

$$\hat{\mathscr{H}}_w = 1 - \frac{2}{k(k-1)} \sum_{j<i\leq l} \left( \frac{\mathbf{W}_{i,:} \cdot \mathbf{W}_{j,:}}{||\mathbf{W}_{j,:}||\,||\mathbf{W}_{j,:}||} \cdot \right) \tag{4}$$

A well trained classifier will have a value of $\hat{\mathscr{H}}_w$ very close to 1.

**Note** that $k$ weight vectors are projecting a $q$-dimension feature vector to a $k$-dimension vector space, for $q > k$. If the weight vectors are orthogonal, their directions define an orthogonal basis in the $k$-dimension space. Because of the isomorphism of the Euclidean space, once the weight vectors of classifier $h(\cdot; \theta_h)$ are (almost) orthogonal their updating is unlikely to improve overall performance of the DNN. We believe, at that point, the focus should be to improve the feature extractor's parameters.

In the next section, we develop a method for evaluation of training quality of the feature extractor $g(\cdot; \theta_g)$. It is a much harder task because feature extractors are, in general, much more complex and *directly quantifying* its parameters with a few metrics is a near impossible task; we develop a method for *indirectly evaluating* the feature extractor without training or testing data.

## 3 TRAINING QUALITY OF THE FEATURE EXTRACTOR $g(\cdot; \theta_g)$

We utilize the given DNN to *synthesize* or *generate* input data vectors that are used to evaluate the feature extractor. We use the terms synthetic or generated interchangeably. We refer to these generated data as *prototypes* to distinguish them from *original* data used for developing the DNN. Our generation methods described next ensures that synthesized data is classified correctly.

A prototype generation starts with an input-data and a target output class. The input (image) is forward passed to generate an output, the output of the DNN and the target class are used to compute loss, the obtained loss is back propagated to the input, and the input image is updated until the loss is below a target threshold. Because the synthesized data is generated utilizing DNN's parameters, it reflects trained DNN's inherent characteristics. Below we describe our proposed data generation methods.

### 3.1 METHODS FOR DATA GENERATION

To generate a prototype iteratively we use the loss function defined by Eqn. 5 (but this is not a limitation because the method can be used with other loss functions as well) and the update rule Eqn. 6, where $t$ is iteration number, $\mathbf{m}_t$ is the current input to the network.

$$\textbf{Loss Function: } \mathscr{L} = -\sum_{j=1}^{k} y_{\mathbf{m}_t, j}\, log(f_j(\mathbf{m}_t)) \tag{5}$$

$$\textbf{Update Rule: } \mathbf{m}_{t+1} \leftarrow \mathbf{m}_t - \eta \nabla_{\mathbf{m}_t} \mathscr{L} \,/||\, \nabla_{\mathbf{m}_t} \mathscr{L} \,|| \tag{6}$$

Let us assume that one-hot encoded output $\mathbf{y}_l$, that is, for a given class $l \in [k]$, $y_l = 1$ and $y_j = 0$ for $j \neq l$. To create a prototype example $\mathbf{m}^{(l)} \in \mathbb{R}^p$ for class $l$, we initialize $\mathbf{m}_0^{(l)}$ with a random vector

$\mathbf{r} \in \mathbb{R}^p$ and forward pass $\mathbf{m}_0^{(l)}$ to compute cross-entropy loss between the output $f(\mathbf{m}_0^{(l)}; \theta)$ and $\mathbf{y}_l$, backpropagate the loss, and update the input vector using rule in Eqn. 6 to obtain $\mathbf{m}_1^{(l)}$. The iterative process continues until observed loss is below a desired (small) value.

**Probabilities for Core Prototypes:** A core prototype for a class $l \in [k]$ should produce ideal output; the output probability $p_l$ of the intended class is (almost) '1' and zero for others.

$$p_l = 1 \text{ and } p_j = 0 \text{ for } j \neq l \tag{7}$$

First, we use the above probability distribution to generate a set of *seed* prototypes, which are then used to generate core prototypes (see description below).

## 3.2 GENERATION OF PROTOTYPE DATASETS

First we generate a set of $k$ *seed* prototypes, exactly one seed for each category. The starting input vector for a seed prototype is a random vector $m_0 \in \mathbb{R}^p$, a target class label $l \in [k]$ and a small loss-value $\delta_{loss}$ for termination of iteration (see Sec. 3.2). We use these seed prototypes to generate $(k-1)$ *core prototypes* for each category.

---

**Algorithm 1** Generate $k$ Seed Prototypes $\mathbf{S}$

---

1: **procedure** GENERATESEEDSPROTOTYPES
2:     Input: $\delta_{loss}$                                     ▷ Loss for iteration termination
3:     Output: $\mathbf{S}$
4:     $\mathbf{S} = \oslash$
5:     for each $l \in k$
6:         $\mathbf{S}_0^{(l)} = \mathbf{r} \in \mathbb{R}^p$, $\mathcal{L} = \infty$, and $t = 0$
7:         while $\mathcal{L} > \delta_{loss}$
8:             $\mathcal{L} = -\sum_{j=1}^{k} y_{\mathbf{S}_t^{(l)}, j} \log(f_j(\mathbf{S}_t^{(l)}))$                 ▷ Eqn. equation 5
9:             $\mathbf{S}_{t+1}^{(l)} = \mathbf{S}_t^{(l)} - \eta \nabla_{\mathbf{S}_t^{(l)}} \mathcal{L} / \|\nabla_{\mathbf{S}_t^{(l)}} \mathcal{L}\|$        ▷ Eqn. equation 6
10:           $t = t + 1$
11:         $\mathbf{S} = \mathbf{S} \bigcup \mathbf{S}_t^{(l)}$
12: **end procedure**

---

**Generation of Seed Prototypes:** Let $\mathbf{S}_0 = \{\mathbf{S}_0^{(l)} | l \in [k]\}$ be the set of $k$ random vectors drawn from $R^p$. Starting with each $\mathbf{S}_0^{(l)}$ generate one seed prototype for class $l$ by iteratively applying update rule given by Eqn. 6 and output probability distribution defined by Eqn. 7. Let the seed vector obtained after termination of iterations that started with initial vector $\mathbf{S}_0^{(l)}$ be denoted by $\mathbf{S}^{(l)}$ and let $\mathbf{S}$ be the collection of all such $k$ vectors. A procedure for generating $n$ seed-prototypes is outlined in the Algorithm 1.

These seed vectors serve two purposes: a *preliminary* evaluation of inter-class separation and *initial* or *starting input* for generating prototypes to characterize intra-class compactness. We generate $(k-1)$ prototypes for each class.

**Core or Saturating Prototypes:** Let $S_{c,i}^{(l)}$ be the prototype for class $l$ generated starting from the seed $S^{(i)}$ of the class $i$. We expect that $S_{c,i}^{(l)}$ to be the closest input data that is within the boundary of the class $l$ and closest to the boundary of the class $i$. It is important to note that output produced by prototype $\mathbf{S}^{(j)}$ satisfies the Eqn. 7. As shown in the Algorithm 2, the generation process initiates a prototype with $\mathbf{S}^{(j)}, j \neq l$ and iterates using update rule (Eqn. 6) until it produces outputs that satisfy Eqn. 7 for the class $l$. Basically, we are starting with a prototype within the polytope of the class $j$ and iteratively updating it until the updated prototype has moved (deep) inside the class $l$. We repeat the process for all $l \in [k]$ to generate $\mathbf{S}_c = \{\mathbf{S}_c^{(1)}, \mathbf{S}_c^{(2)}, \cdots, \mathbf{S}_c^{(k)}\}$. Since the output produced by these prototypes create saturated outputs, we call the prototypes in $\mathbf{S}_c$ *core* or *saturating* prototypes.

---

**Algorithm 2** Generate $k(k-1)$ Core Prototypes $\mathbf{S}_C$

---

1: **procedure** GENERATECOREPROTOTYPES
2:     Input: $\delta_{loss}$ and $S$                     ▷ Loss for iteration termination and $k$ seed prototypes
3:     Output: $\mathbf{S}_c$                     ▷ $S_c$ will have $k(k-1)$ core prototypes, $(k-1)$ for each class
4:     $\mathbf{S}_c = \oslash$
5:     for each $l \in k$
6:         for each $(j \in k)$ and $j \neq l$                     ▷ Find $(k-1)$ prototypes for class $l$
7:             $\mathbf{S}_{c,0}^{(j)} = \mathbf{S}^{(j)}$, $\mathscr{L} = \infty$, and $t = 0$                     ▷ Start from the seed prototype in the class $j$
8:             while $\mathscr{L} > \delta_{loss}$
9:                 Compute $\mathscr{L}$ using probability distribution given by Eqn. equation 7 for class $l$
10:                 $\mathbf{S}_{c,t+1}^{(j)} = \mathbf{S}_{c,t}^{(j)} - \eta \nabla_{\mathbf{S}_{c,t}^{(j)}} \mathscr{L} / \| \nabla_{\mathbf{S}_{c,t}^{(j)}} \mathscr{L} \|$                     ▷ Eqn. equation 6
11:                 $t = t+1$
12:         $\mathbf{S}_c = \mathbf{S}_c \bigcup \mathbf{S}_{c,l}^{(j)}$
13: **end procedure**

---

The prototypes generated using two algorithms described in this section are used for evaluation of the quality of the feature extractor.

### 3.3   METHOD FOR EVALUATION OF THE FEATURE EXTRACTOR

We use $g(\cdot; \theta_g)$ to extract the features of the generated prototypes. Let $V_{c,i}^{(l)}$ be the feature vector generated from the seed $S_{c,i}^{(l)}$, that is, $V_{c,i}^{(l)} = g(S_{c,i}^{(l)}; \theta_g)$. Let all feature vectors produced from the seed vectors of the prototypes of the class $l$ be denoted by $V_c^{(l)} = \{V_{c,1}^{(l)}, V_{c,1}^{(l)}, \cdots, V_{c,k}^{(l)}\}$.

**Definition 3.1** (Cosine Similarity). *Given two vectors* $\mathbf{v}_1$ *and* $\mathbf{v}_2$ *of same dimension, cosine similarity*

$$CosSim(\mathbf{v}_1, \mathbf{v}_2)) = \cos(\theta) \equiv (\mathbf{v}_1 \cdot \mathbf{v}_2)/(\|\mathbf{v}_1\|\|\mathbf{v}_2\|) \tag{8}$$

It is obvious that $CosSim(\mathbf{v}_1, \mathbf{v}_2))$ of $\mathbf{v}_1$ and $\mathbf{v}_1$ is the *dot* product of their unit vectors. We normalize all vectors in $V_c^{(l)}$ to obtain $v_c^{(l)}$ for computing all-pair *CosSim* of the vectors in $V_c^{(l)}$. For our set of $k$ vectors in each class, we need to compute $k(k-1)/2$ *CoSim* since that is the number of possible pairwise combinations. Matrix multiplication provides a convenient method for computing these values. Let $\mathbf{v}_{c,i}^{(l)}$ be the $i$th row of the matrix $\mathbf{G}^{(l)}$ and let $(G^{(l)})^T$ be the transpose of $G^{(l)}$. The elements of the $k \times k$ matrix $(\mathbf{G}^{(l)}(G^{(l)})^T)$ are $(\mathbf{v}_{c,i}(l) \cdot \mathbf{v}_{c,j}(l))$, the $CosSim(\mathbf{v}_{c,i}^{(l)}, \mathbf{v}_{c,j}(l))$. ach diagonal element, being *dot* products of a unit vector with itself, is 'one' and also, the $(\mathbf{G}^{(l)}(G^{(l)})^T)_{i,j} = (\mathbf{G}^{(l)}(G^{(l)})^T)j, i$, we need to consider only off-diagonal the upper or lower triangular part of the matrix.

**Intra-class or within-class Similarity Metric,**   $\mathscr{M}_{in}$, postulates *intra-* or *within*-class prototypes of a well trained DNN are very similar for a given class and their pairwise *CosSim* values are close to *one*. Since our prototype generation algorithms generate $k$ sets of prototypes (one set for each class) for a $k$-class classifier, it is important to evaluate within-class similarity of all prototypes. To get an estimate of within-class similarity of a DNN, estimates of all classes are combined.

**Definition 3.2** (Estimate of intra- or within-class similarity metric). *An estimate of intra- or within-class similarity metric* $\mathscr{M}_{in}$ *is given by*

$$\mathscr{M}_{in} = \frac{1}{k} \sum_{l \in k} \left( \frac{2}{k(k-1)} \sum_{j < i \leq k} ((\mathbf{G}^{(l)}(G^{(l)})^T)_{i,j}) \right) \tag{9}$$

The variance of $\mathscr{M}_{in}$ of a well trained DNN should be very low as well. Our evaluations reported in Sec. 4 show that $\mathscr{M}_{in}$ for CIFAR10 is above 0.972 even when it is trained with 25% of the trained data and variance is also small, an indication that the DNN was probably trained well. But for CIFAR100 $\mathscr{M}_{in}$ and variance are relatively higher, indicating that it was not trained well. **We propose to use within-class similarity metric to define upper bound for test accuracy** (see Sec. 4).

**Inter-class or between-class Separation Metric,** $\mathscr{M}_{bt}$, postulates that two prototypes of different classes should be less similar and their pair-wise *CosSim* value is close to *zero*. For making $\mathscr{M}_{bt}$ increase towards one as the quality of a trained DNN increases, we define $\mathscr{M}_{bt} = 1 - CosSim(\mathbf{v}_{c,i}^{(l_1)}, \mathbf{v}_{c,j}^{(l_2)})$, where $\mathbf{v}_{c,j}^{(l_1)}$ and $\mathbf{v}_{c,j}^{(l_2)}$ are from class $l_1$ and $l_2$, respectively.

Because every class has $k$ prototypes and estimation of $\mathscr{M}_{bt}$ requires further considerations. Consider two classes $l_1$ and $l_2$, we can pick one prototype from class $l_1$ and compare it with $k$ prototypes in $l_2$. This process is repeated for all prototypes in the class $l_1$ requiring $k^2$ *CosSim* computations; also, since each prototype in class $l_1$ must be compared with all prototypes in other $(k-1)$ classes, total $(k-1)k^2$ comparisons are necessary. Overall total computational complexity is $O(k^4)$. If computation time is an issue, one can get a reasonable estimate with $O(k^3)$ computations, which is described next.

Let the core prototypes $S_{c,l}^{(2)}, S_{c,l}^{(3)}, \cdots, S_{c,l}^{(k)}$ be generated from seed prototype $S^{(l)}$ for $l \in [k]$. From these prototypes, we extract corresponding feature vectors, and normalize them to get $v^{(1)}, v_{c,1}^{(2)}, v_{c,1}^{(3)}, \cdots, v_{c,l}^{(k)}$. Similar to matrix $G$, we define a matrix $H$ from these normalized feature vectors and use it to estimate inter-class separation metric.

**Definition 3.3** (Estimate of inter- or between-class separation metric)**.** *An estimate of inter- or between-class separation metric $\mathscr{M}_{bt}$ is given by*

$$\hat{\mathscr{M}}_{bt} = 1 - \frac{1}{k} \sum_{l \in k} \left( \frac{2}{k(k-1)} \sum_{j < i \le k} ((\mathbf{H}^{(l)}(H^{(l)})^T)_{i,j}) \right) \tag{10}$$

Because each element of the matrix $HH^T$ is the cosine similarities (see Eqn. 8) of a pair of features of prototypes from different classes, $HH^T \in [0,1]^{k \times k}$. In Sec. 4, **we will use values of $\mathscr{M}_{bt}$ and their variances to define a lower bound for test accuracy.**

### 3.4 Evaluation of Training Quality of Feature Extractor

When a feature classifier's weights are pairwise orthogonal (our observations support this, see Table 1), overall classification accuracy will depend on the features that the feature extractor generates. If feature vectors of all testing examples of a class are similar and their standard deviation is very low, the value of $\hat{\mathscr{M}}_{in}$ will be high; but features of the between-class are very dissimilar and their standard deviation is high, the value of $\hat{\mathscr{M}}_{bt}$ will be low. This situation will cause classification errors. We believe during training time within-class feature vectors' become very close to each other faster and will represent an upper bound for testing accuracy.

Also, we believe it is much harder to impart training so that feature vectors of one class move far away from that of another class. This will make the value of the metric $\hat{\mathscr{M}}_{bt}$ lower. And this will act as a lower bound for the testing accuracy. Our observations empirically support this hypothesis.

## 4 Empirical Evaluation of Proposed Metrics

**Datasets** We have used the image classification datasets CIFAR10 (Krizhevsky et al., a), and CIFAR100 (Krizhevsky et al., a). All of these datasets contain labeled 3-channel 32x32 pixel color images of natural objects such as animals and transportation vehicles. CIFAR10 and CIFAR100 have 10 and 100 categories/classes, respectively. The number of training examples per category are 5000 and 500 for CIFAR10 and 500, respectively. *We created 7 datasets from each original dataset by randomly selecting 25%, 40%, 60%, 70%, 80%, 90%, and 100% of the data.*

**Training** For results reported we have used ResNet18 (He et al., 2015), defining $g(\cdot; \theta_g)$ as the input to the flattened layer (having 256 neurons) after global average pooling and $h(\cdot; \theta_h)$ as the fully connected and softmax layers. All evaluations were completed on a single GPU.

For each dataset, we randomly initialize a ResNet18 (He et al., 2015) network and train it for a total of 200 epochs. The learning rate starts at 0.1 in the first 100 epochs, and then it is reduced to 0.05 for the last 100 epochs. For each data set we trained 35 networks, 7 partitioned datasets and 5 randomly initialized weights to reduce biases.

## 4.1 Evaluation of the Classifier

With the network frozen, values of the metric $\hat{\mathbf{H}}_w$ were calculated. Because the values were so close to one we converted them into angles and they are summarized in Table 1. It shows that the weight vectors are almost orthogonal even when only 25% of the data is used.

| Taring Data % | 25 | 40 | 60 | 70 | 80 | 90 | 100 |
|---|---|---|---|---|---|---|---|
| CIFAR10 | 89.98 | 89.99 | 89.99 | 89.99 | 89.99 | 89.99 | 89.99 |
| CIFAR100 | 89.99 | 89.99 | 89.99 | 89.99 | 89.99 | 89.99 | 89.99 |

Table 1: Mean Angles in Degrees Between Weight Vectors $\mathbf{W}$ of the Classifier $h(\cdot; \theta_h)$.

## 4.2 Evaluation of the Feature Extractor

For evaluating the feature extractor, we used the frozen network to generate seed prototype images using Algorithm 1 with a learning rate of 0.1 for CIFAR100 and 0.01 for CIFAR10, respectively. Then Algorithm 2 was used to generate $(k-1)$ *core* prototypes (see Sec.3.2) for each category. Thus, for CIFAR10 we generate a total of 100 prototypes and for CIFAR100 we generate a total of 10,000 prototypes. The process was repeated five times to eliminate random selection bias and reported results are averages of these five runs. It is worth mentioning that we examined the data from each run and found no glaring differences.

Our observations are summarized in Table 2. The top-half of the table shows results for the CIFAR10 dataset and its fractions. The 7 rows are for the 7 datasets we created from the original dataset by partitioning it. The 1st column shows percent of data used. The second and third columns show values of the mean and standard deviation of the within-class similarity measure $\hat{\mathcal{M}}_{in}$.

**An Upper Bound for Testing Dataset Accuracy:** The next column shows adjusted values of $\hat{\mathcal{M}}_{in}$, obtained by subtracting two standard deviations for increasing confidence of the observations. *These values give us an upper bound for accuracy that one would have observed from the testing dataset.* As discussed before, for a given class as the value of $\hat{\mathcal{M}}_{in}$ increases and its standard deviation decreases, the features of the class are close to each other and test data performance should be high.

| Taring Data % | Mean within CosSim $\hat{\mathcal{M}}_{in}$ | Within CosSIn Std (inStd) | ($\hat{\mathcal{M}}_{in}$ - 2*Std) $\hat{\mathcal{M}}_{in}$ | Mean between CosSim (CSbt) | Between CosSim Std (CSbtStd) | CSbt +2btStd | $\hat{\mathcal{M}}_{bt}$ | Accuracy |
|---|---|---|---|---|---|---|---|---|
| Test Accuracy Bounds for ResNet14 trained with CIFAR10 (256 feature-layer neurons) | | | | | | | | |
| 25 | 0.9727 | 0.0129 | 0.9470 | 0.2354 | 0.3922 | 0.6002 | 0.6078 | 0.7099 |
| 40 | 0.9793 | 0.0093 | 0.9607 | 0.2511 | 0.3975 | 0.5807 | 0.6025 | 0.7675 |
| 60 | 0.9825 | 0.0087 | 0.9652 | 0.2277 | 0.3585 | 0.5292 | 0.6415 | 0.8149 |
| 70 | 0.9837 | 0.0082 | 0.9673 | 0.2218 | 0.3487 | 0.5168 | 0.6513 | 0.8249 |
| 80 | 0.9824 | 0.0088 | 0.9647 | 0.2339 | 0.3697 | 0.5113 | 0.6303 | 0.8362 |
| 90 | 0.9849 | 0.0072 | 0.9704 | 0.2269 | 0.3631 | 0.4900 | 0.6369 | 0.8449 |
| 100 | 0.9844 | 0.0077 | 0.9690 | 0.2292 | 0.3690 | 0.4614 | 0.6310 | 0.8548 |
| Test Accuracy Bounds for ResNet18 trained with CIFAR100 (256 feature-layer neurons) | | | | | | | | |
| 25 | 0.9134 | 0.0285 | 0.8565 | 0.3721 | 0.1140 | 0.6002 | 0.3998 | 0.3256 |
| 40 | 0.9168 | 0.02523 | 0.8664 | 0.3656 | 0.1075 | 0.5807 | 0.4193 | 0.414 |
| 60 | 0.9233 | 0.0229 | 0.8776 | 0.3337 | 0.0977 | 0.5292 | 0.4708 | 0.4982 |
| 70 | 0.9248 | 0.0221 | 0.8807 | 0.3269 | 0.0949 | 0.5168 | 0.4832 | 0.5282 |
| 80 | 0.9252 | 0.0218 | 0.8815 | 0.3267 | 0.0923 | 0.5113 | 0.4887 | 0.5526 |
| 90 | 0.9273 | 0.0216 | 0.8842 | 0.3112 | 0.0894 | 0.4900 | 0.5100 | 0.5685 |
| 100 | 0.9331 | 0.0204 | 0.8923 | 0.2897 | 0.0858 | 0.4614 | 0.5386 | 0.5945 |

Table 2: ResNet18 trained with CIFAR10 and CIFAR100 datasets (256 neurons at the feature layer).

Let us explore the columns 5 to 8 of the Table 2. Values in the columns 5 to 7 are analogous to the values in the values in the columns 2 to 4, but for between-class similarity measures. For a well

trained network these values should be small. Calculation of values in column 4 differs from that in column 7 because it is obtained by adding (instead of subtracting) two standard deviations for increasing confidence of the observations.

**A Lower Bound for Test Dataset Accuracy:** The column 8 is obtained after subtracting the values in column 7 (see Eqn. 10). *These values give us a lower bound for accuracy one would get from the testing dataset.* As discussed earlier, for a given class as the value of $\hat{\mathcal{M}}_{bt}$ increases and its standard deviation decreases, the features of the class are much different from those of other classes and accuracy observed from testing dataset should increase.

The data in the *column 9 shows accuracy* of the trained networks obtained *when tested with test dataset.* Note that we used all data in the test dataset (not a fraction as used for the training of the network). For ease of comparing and contrasting, the values of the Upper and Lower Bounds as well as test Accuracy for CIFAR10 is shown in Fig. 1. We can see that observed accuracy is correctly bounded by calculated values of the metrics.

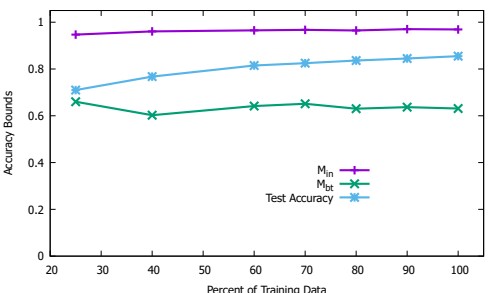 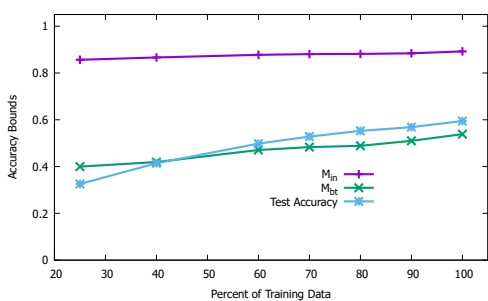

Figure 1: CIFAR 100 Accuracy Bounds      Figure 2: CIFAR 100 Accuracy Bounds

The lower half of the Table 2 and the Fig. 2 is for CIFAR100 dataset. Except for 25% and 40% of the training data the bounds obtained from the network have enclosed the accuracy correctly. But a closer examination of the data in the table, tells us that 25% and 40% data created a very poorly trained network. Thus, it may not be worth using them for any serious applications.

## 5   CONCLUSION

In this paper we have proposed and evaluated a method for dataless evaluation of trained DNN that uses one-hot coding at the output layer, and usually implemented using softmax function. We assumed that the network is a composition of a feature extractor and a classifier. The classifier is a one-layer fully-connected neural network that classifies features produced by the feature extractor. We have shown that the weight vectors of trained classifier-layer are (almost) orthogonal. Our empirical evaluations have shown that the orthogonality is achieved very quickly even with a small amount of data and performance of the overall DNN is quite poor.

The feature extractor part of a DNN is very complex and has a very large number ( usually many millions) of parameters. We believe a direct evaluation of the parameters of the feature extractor is an extremely difficult, if not impossible, task. We have proposed two metrics for indirect evaluation of feature extractors. Using these metrics we were able to define an upper and lower bound for classification accuracy of a trained network. It is important to note that our posed method finds values of these metrics from a given DNN without any knowledge of training and evaluation data.

While empirical evaluations show that the bounds enclose the *real* test accuracy obtained from the test dataset, we are working to tighten the gap between them. We believe the bounds can be used to define loss-function for the feature extraction layer for developing classifiers that discourages overfilling and improves performance of the trained network. We have developed a loss function and are currently under evaluation.

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
