# OpenReview forum: "Fantastic DNN-Classifier Identification without Testing Dataset"
_ICLR.cc/2024/Conference — Submitted to ICLR 2024_

### Official Review · Reviewer_THQj · 2023-10-18

**Soundness:** 2 fair
**Presentation:** 2 fair
**Contribution:** 3 good
**Rating:** 5
**Confidence:** 3

**Summary:**

This paper proposes an approach to estimate the bounds for test accuracy of a given model without access to test data. In particular,it first generates prototype vectors for each class, and then generates k-1 additional vectors for each initial prototype (by “perturbing” the initial prototype towards the class boundary with respect to each of the other k-1 classes). The aggregated pairwise cosine similarity of the prototypes (intra-class and inter-class) is used to estimate the test accuracy. To evaluate their methods, the authors show their method can successfully bound models’s test accuracy on CIFAR-10 and CIFAR-100.

**Strengths:**

1.an interesting problem to study

2.smart, intuitive methodology

3.the proposed method seems to work fine for bounding testing accuracy

**Weaknesses:**

1.insufficient evaluation

The evaluation is only conducted on two similar ResNet models and two small datasets (CIFAR-10 and CIFAR-100) and makes the results less convincing. At least an additional model on a larger dataset (e.g., ImageNet) should be checked.

2.writing can be improved

The language usage is sub-optimal. For example, there are five “We believe” in the paper which should not be used frequently. There are also quite a few grammar errors: e.g., at the end of introduction, “The metric for assessing the classifier, we use its weight vectors”.

**Questions:**

-table1, what if less than 25% data is used?

-how generalizable the proposed method is on a larger dataset?

---

### Official Review · Reviewer_Szfc · 2023-10-31

**Soundness:** 2 fair
**Presentation:** 2 fair
**Contribution:** 2 fair
**Rating:** 3
**Confidence:** 4

**Summary:**

The paper creatively tried to evaluate trained deep neural networks without any data, either training, validation or testing data. The idea of evaluating the network with no data is innovative and attractive, however, the empirical results are not convincing enough that the approach is ready to be used.

**Strengths:**

1. the idea of evaluating the network without any data is very attractive.

2. the overall flow of the paper is written in a fairly straightforward way, easy to follow, although there are many typos.

**Weaknesses:**

1. the major issue is the application of this method. The authors unfortunately didn't show convincing evidence that the method is applicable to a broader scope. For example, the experiments are only conducted with ResNet18 over CIFAR10 and CIFAR100 datasets, the choices such as learning rates are from a fixed pattern, so it's hard to evaluate how generalize this method is.
    - in addition, the ResNet18 over CIFAR10 dataset can achieve over 90 percent accuracy since years ago, e.g. see https://github.com/kuangliu/pytorch-cifar the authors can only get 85 does not seem very convincing.

2. the results at Table 1 are all the same number, this also seems weird.

3. there are quite a few typos, e.g., ResNet14, CIFAR10 and 500, etc. In eq. (9), there are two types of G, and in (10), there are two different types of H, which posts challenge of reading this paper further.

**Questions:**

1. it could be helpful if the authors explain the reason why numbers in table 1 are all the same.

---

### Official Review · Reviewer_QA6N · 2023-11-01

**Soundness:** 2 fair
**Presentation:** 1 poor
**Contribution:** 2 fair
**Rating:** 3
**Confidence:** 4

**Summary:**

This paper proposes a novel method for estimating the test accuracy of a DNN classifier without any test data. Specifically, the paper proposes three metrics to assess the training quality of the classification layer and the feature extraction backbones. The proposed method is evaluated with two models (Resnet-14 and Resnet-18) on two datasets (Cifar-10 and Cifar-100).

**Strengths:**

- A novel method for estimating the test accuracy of a DNN classifier.
- Empirical evaluation with two datasets and two models.

**Weaknesses:**

- Significance of the paper.
I felt confused about the significance of the proposed research when I read the abstract-what is the importance of estimating a DNN classifier’s accuracy without any test data? Unfortunately, I did not find a clear motivation after reading the introduction. The test accuracy of a DNN is strongly related to the dataset it is going to be tested. Without any test data, one may only estimate its accuracy on the training dataset or a test dataset from a similar distribution. However, in the real-world scenario, practitioners usually found the test data shifted from the training data. In this case, the proposed research seems not to be significant enough without further clarification and discussion.
Besides, clearly, the proposed method may only work for DNN classifiers. Therefore, I would suggest replacing the terms of general DNN with DNN classifiers throughout the paper.

- Presentation is not good enough.
I believe the paper could be significantly improved with another round of revision. The introduction does not provide a broader assessment of the importance of the research topic while introducing some very basic details about training a DNN and details of the proposed method. Besides, the related work is extremely short and may make the audience confused about whether the proposed research problem is important enough.

- Missing related work.
The references only include 19 papers, and 2 of them are datasets. The review of the relevant work is simply not comprehensive and detailed enough.
There is an important line of related work that uses a few test examples to estimate the test accuracy of a model [1] [2] [3] [4]. I feel it is necessary to include them as that would help audiences understand why estimating test accuracy without any test data is important compared with sample-efficient testing work.

[1] Kossen, Jannik, et al. “Active testing: Sample-efficient model evaluation.” International Conference on Machine Learning. PMLR, 2021.

[2] Kossen, Jannik, et al. “Active surrogate estimators: An active learning approach to label-efficient model evaluation.” Advances in Neural Information Processing Systems 35 (2022): 24557-24570.

[3] Chen, Junjie, et al. “Practical accuracy estimation for efficient deep neural network testing.” ACM Transactions on Software Engineering and Methodology (TOSEM) 29.4 (2020): 1-35.

[4] Li, Zenan, et al. “Boosting operational DNN testing efficiency through conditioning.” Proceedings of the 2019 27th ACM Joint Meeting on European Software Engineering Conference and Symposium on the Foundations of Software Engineering. 2019.

**Questions:**

How does the proposed method compare with the sample-efficient test accuracy estimation methods? What is the importance and significance of estimating a DNN’s accuracy without any test data?

---

### Meta-Review · Area_Chair_XNTW · 2023-12-02

**Metareview:**

The paper proposes estimating the test accuracy of a DNN classifier without any test data. The main method is to generate class-wise prototype vectors, perturb them, and estimate pairwise cosine similarities. The experiments are done on Cifar datasets to estimate the bounds of test accuracies.

**Justification For Why Not Higher Score:**

Lack of enough motivation for the problem in the Introduction.

Lack of comprehensive review of previous work.

Limited experimental evaluations, e.g., only on standard Cifar datasets and ResNets.

Test accuracies of the evaluated settings seem lower than standard.

The paper writing can be improved.

**Justification For Why Not Lower Score:**

N/A

---

### Decision · Program_Chairs · 2024-01-16

Reject